# Toxicity of Carlina Oxide—A Natural Polyacetylene from the *Carlina acaulis* Roots—In Vitro and in Vivo Study

**DOI:** 10.3390/toxins12040239

**Published:** 2020-04-09

**Authors:** Artur Wnorowski, Sylwia Wnorowska, Kamila Wojas-Krawczyk, Anna Grenda, Michał Staniak, Agnieszka Michalak, Sylwia Woźniak, Dariusz Matosiuk, Grażyna Biała, Magdalena Wójciak, Ireneusz Sowa, Paweł Krawczyk, Maciej Strzemski

**Affiliations:** 1Department of Biopharmacy, Medical University of Lublin, 20-093 Lublin, Poland; artur.wnorowski@umlub.pl (A.W.); sylwiaw211@gmail.com (S.W.); 2Department of Pneumology, Oncology and Allergology, Medical University of Lublin, 20-090 Lublin, Poland; kamila.wojas-krawczyk@umlub.pl; 3Laboratory for Immunology and Genetics, Medical University of Lublin, 20-093 Lublin, Poland; anna.grenda@umlub.pl (A.G.); pawel.krawczyk@umlub.pl (P.K.); 4Department of Analytical Chemistry, Medical University of Lublin, 20-093 Lublin, Poland; michal_staniak@wp.pl (M.S.); kosiorma@wp.pl (M.W.); i.sowa@umlub.pl (I.S.); 5Chair and Department of Pharmacology and Pharmacodynamics, Medical University of Lublin, 20-093 Lublin, Poland; agnieszka.michalak@umlub.pl (A.M.); grazyna.biala@umlub.pl (G.B.); 6Chair and Department of Synthesis and Chemical Technology of Pharmaceutical Substances, Medical University of Lublin, 20-093 Lublin, Poland; sylwia.wozniak@umlub.pl (S.W.); dariusz.matosiuk@umlub.pl (D.M.)

**Keywords:** cytotoxicity, phytotherapy, folk medicine, zebra fish, PD-L1, carlina oxide

## Abstract

There are several reports indicating that the roots of the *Carlina acaulis* L. used to be commonly applied as a treatment measure in skin diseases and as an antiparasitic agent, starting from antiquity to the 19th century; however, nowadays, it has lost its importance. Currently, numerous studies are being conducted assessing the possibility of reintroducing *C. acaulis*-derived extracts to phytotherapy. Determining the safety profile of the main constituents of the plant material is crucial for achieving this goal. Here, we aimed to determine the toxicity profile of carlina oxide, one of the most abundant components of the *C. acaulis* root extract. We obtained the carlina oxide by distillation of *C. acaulis* roots in the Deryng apparatus. The purity of the standard was evaluated using GC-MS, and the identity was confirmed by IR, Raman, and NMR spectroscopy. In vitro cytotoxicity was assessed using a panel of human cell lines of skin origin, including BJ normal fibroblasts and UACC-903, UACC-647, and C32 melanoma cells. This was accompanied by an in vivo zebrafish acute toxicity test (ZFET). In vitro studies showed a toxic effect of carlina oxide, as demonstrated by an induction of apoptosis and necrosis in both normal and melanoma cells. Decreased expression of AKT kinase and extracellular signal-regulated kinase 1/2 (ERK1/2) was noted in the UACC-647 melanoma cell line. It was also observed that carlina oxide modified the expression of programmed cell death-ligand 1 (PD-L1) in tested cell lines. Carlina oxide exhibited high in vivo toxicity, with LC_50_ = 10.13 µg/mL upon the 96 h of exposure in the ZFET test. Here, we demonstrate that carlina oxide displays toxic effects to cells in culture and to living organisms. The data indicate that *C. acaulis*-based extracts considered for therapeutic use should be completely deprived of carlina oxide.

## 1. Introduction

*Carlina acaulis* L. is a monocarpic perennial herb from the Asteraceae family. The plant used to be widely recognized in ancient and medieval medicine. Its root used to be applied for treatment of various skin diseases, as well as a diuretic and diaphoretic agent. However, at the end of the 19th century, the medicinal use of the *C. acaulis* root ceased. It is not clear why this raw material was withdrawn from medical practice; perhaps its importance was lost due to either lack of availability, limited effectiveness, or undesirable toxicity. The literature provides limited data on the adverse effects related to *C. acaulis*. Nevertheless, it has been documented that this raw material was used as an anthelmintic, and is still of importance in folk medicine [1]. In addition, emetic [2,3] and abortive effects [2] have been reported.

The roots of *C. acaulis* are rich in inulin and chlorogenic acids [4,5]. They also contain an essential oil (1–2%) [4,6]. Carlina oxide, 2-(3-phenylprop-1-ynyl)furan, is a natural polyacetylene that constitutes up to 90–99% of the essential oil [1,7]. Pure carlina oxide isolated from *C. acaulis* has been reported to be toxic to larvae of *Culex quinquefasciatus* [8], cultured *Trypanosoma brucei* cells, and several strains of microbial agents [9]. However, *C. acaulis* root extracts devoid of carlina oxide have displayed no cytotoxicity to human cells in vitro. Additionally, carlina oxide-free extracts significantly stimulated the proliferation of skin cells of human origin [10]. Thus, we hypothesized that toxic properties of the *C. acaulis* root and its extracts depend on the activity of carlina oxide.

Our research objective was to comprehensively assess the toxicity of *C. acaulis*-derived carlina oxide in in vitro and in vivo models. We aimed to address the safety of *C. acaulis* root preparations used in folk medicine and evaluate the possibility of reintroducing this plant into phytotherapy.

## 2. Results

### 2.1. The Identity and Purity of Carlina Oxide

Chromatographic analysis showed the presence of one major component of the oil. On the basis of the retention index, molecular ion mass, and spectroscopic analyses, it was found that this compound is carlina oxide. Its percentage content in the tested oil was 96.2%. The oil also contained small amounts of benzaldehyde (0.57%), *ar*-curcumene (0.56%), and *β*-sesquiphellandrene (0.16%). The chromatographic and mass data are presented in Appendix A. The total ion chromatogram, MS, IR, Raman, and NMR spectra are presented in Appendix A.

### 2.2. Carlina Oxide Induces Cell Death in Cultured Cells

An initial cytotoxicity assessment of carlina oxide was conducted using a panel of normal and cancerous skin cells of human origin. BJ normal foreskin fibroblasts and three melanoma cell lines (UACC-903, UACC-647, and C32) were subjected to increasing concentrations of carlina oxide (Figure 1). Doses ranged from 3.125 µg/mL (17 µM) to 50 µg/mL (274 µM). Assessed lines responded differently to the test compound. The number of viable BJ cells significantly decreased at doses of carlina oxide as low as 3.125 µg/mL. This was accompanied by an increase in the number of necrotic and early apoptotic cells. The viability of UACC-647 melanoma cells also dropped upon carlina oxide treatment. This was followed by an increase in the proportion of late apoptotic cells. UACC-903 and C32 were largely resistant to carlina oxide treatment. Representative images of the tested cells are presented in Appendix A.

### 2.3. Effects of Carlina Oxide on the Expression of Toxicity and Immunological Markers

In order to get more insight into the molecular mechanisms driving proapoptotic activity of carlina oxide, we conducted immunoblotting experiments. The UACC-647 cell line was selected for testing, as it showed the strongest response among the evaluated cell lines. Carlina oxide (50 µg/mL) decreased the expression of AKT and extracellular signal-regulated kinase 1/2 (ERK1/2), the key signaling nodes driving proliferation and cell survival (Figure 2A). There was no apparent change in the expression of eukaryotic elongation factor 2 (eEF2), nor the proliferating cell nuclear antigen (PCNA) (Figure 2B). Expression of β-actin was stable across treatments.

PD-L1 is one of the immune checkpoints [11]. Certain types of tumor cells, including melanoma, are able to express this molecule on their surface. Blocking PD-L1 is one of the most effective anti-cancer treatments. It leads to the activation of the immune system towards cancer cells previously evading the immune response [12,13]. Here, we assessed the ability of carlina oxide to induce PD-L1 expression (Figure 3A). BJ fibroblasts showed significant upregulation of PD-L1 even at the lowest dose used (3.125 µg/mL). A higher dose (50 µg/mL) of carlina oxide was required to increase PD-L1 in UACC-903 and UACC-647 melanoma cells. We were unable to detect PD-L1 mRNA in C32 cells.

Expression of PD-L1 is regulated by microRNA. Disruption of miRNAs complementary to PD-L1 mRNA may affect the efficacy of inhibitors of immunological control points. Thus, we investigated changes in the expression pattern of a panel of miRNA complementary to the PD-L1 mRNA (Figure 3B). However, only minute changes in the expression of miRNA were detected. Concomitantly, we observed no correlation between the expression of any assessed miRNA and PD-L1 expression in BJ, UACC-903, and UACC-647 cells.

### 2.4. Carlina Oxide is Teratogenic to Zebrafish Embryos

An in vitro cytotoxicity assessment was followed by an in vivo toxicity test in zebrafish embryos. Overall survival of embryos in both the negative control and the solvent control was equal to 95.83% at the end of the test. The reliability of the experimental setup was validated using a positive control (acetone, 20 µg/mL), which exhibited the mortality of 33.33% at the end of the 96 h exposure. Altogether, 78 embryos treated with carlina oxide were found dead as a result of coagulation within the first 24 h of the test. Two more larvae were recorded dead at the last time point of the test (i.e., 96 h), as no heartbeat was observed during the 1 min observation (Table 1). The LC_50_ was determined to be 10.13 µg/mL after 96 h of exposure (Figure 4A). Heart function was assessed in living embryos at the end of exposure. Carlina oxide decreased a heartbeat of embryos by 12.5 beats per minute at a dose of 9.375 µg/mL in comparison to control (*p* < 0.05; Figure 4B). Altogether, the results indicated that carlina oxide showed high acute toxicity, as well as the cardiodepressive effects in zebrafish larvae. Moreover, marked developmental changes were observed in 96 hpf (hours post fertilization) in larval zebrafish subjected to 25 µg/mL of the tested compound. The presence of craniofacial malformation (CFM), yolk sac edema (YSE), and shortened tail (ST) indicates teratogenic effects of carlina oxide (Figure 4C).

## 3. Discussion

Formerly, roots of *C. acaulis* were widely used in medicine; however, nowadays, their use is narrowed only for folk medicine. Current research proves that this material contains many compounds of valuable biological properties, and it seems reasonable to reintroduce it to phytotherapy. Toxicity assessment of this raw material is the direction of priority for further research [1].

We conducted a panel of tests to assess the toxicity profile of carlina oxide. In vitro assessment of cytotoxicity towards normal (fibroblasts) and cancer (melanoma) human cells revealed that carlina oxide induces cell death in a cell line-specific fashion. BJ fibroblasts and UACC-647 melanoma cells were susceptible to carlina oxide, although with a different mode of response. Loss of viability of fibroblasts was linked to an increase in necrosis, while in UACC-647 cells, late apoptosis was more prevalent. On the contrary, UACC-903 and C32 melanoma cells were largely resistant to carlina oxide, and no or little suppression of viability was observed in these cells. This selectivity may indicate target-specific toxicity. Presence of a particular receptor or metabolizing enzyme may be necessary for the toxic effect to occur. Melanoma cell lines differ dramatically in terms of gene expression, mutations, motility, invasiveness, and vasculogenic mimicry [14]. This heterogeneity of melanoma cell lines may be reflected in different responses to carlina oxide presented in our study. Herrmann and co-workers studied cytotoxicity of carlina oxide in the HeLa cervical cancer cell line [9]. HeLa cells showed striking resistance to carlina oxide. The cells displayed no change in viability at around 50 µg/mL of carlina oxide, which is comparable to C32 cells in this study. However, HeLa cells survived the exposure of much higher doses of carlina oxide, yielding a LC_50_ value of 446 µg/mL. Pavela and colleagues investigated the effect of *C. acaulis* root essential oil on the viability of normal NHF-A12 human fibroblasts [15]. They observed loss of cell survival of about 40% at a dose of 5 µg/mL after 6 h of treatment. The cytotoxicity was time-dependent; a decrease in the count of viable cells became significant at doses as low as 0.5 and 1 µg/mL upon 24 h of exposure [15]. This is consistent with our data from BJ fibroblasts which displayed an increase in the number of necrotic cells in response to 24 h exposure with 3.125 µg/mL of carlina oxide. Susceptibility of normal human cells to carlina oxide exposure raises safety concerns regarding the usage of *C. acaulis* essential oil in phytomedicine and in food supplements. As mentioned earlier, *C. acaulis* is not commonly used. Yet, its essential oil appeared on the list of botanical food supplements established by Belgium, France and Italy (i.e. BELFRIT list) [16], and carlina oxide was detected in a commercially available multicomponent herbal mixture known as “Swedish herbs” [6]. Although a full toxicological assessment of *C. acaulis* essential oil and carlina oxide is required to draw final conclusions regarding their further use in foods, current data support caution and indicate the need for restriction.

Further evaluation revealed that carlina oxide suppresses the expression of AKT and ERK1/2, signalling nodes related to proliferation and cell survival, and induces the expression of PD-L1. Carlina oxide induces cytotoxicity of tumor cells that can stimulate an immunological response. Neoantigens from necrotic or apoptotic tumor cells may specifically stimulate T lymphocytes, which could destroy tumor cells. On the other hand, damaged tumor cells can express a lot of molecules with an ability to stimulate or inhibit immune cells. One of these molecules is PD-L1, responsible for exhaustion of T lymphocytes in a tumor environment. This explains why we are simultaneously investigating the cytotoxicity and immunological properties of carlina oxide.

The higher the percentage of PD-L1-positive cells, the better response to therapy is expected. However, in some patients, despite a high percentage of PD-L1 positive cells, the response is unsatisfactory, while in other patients, the PD-L1 is low, or there is no expression that is satisfactory and good. This indicates the need to search for additional predictive factors in immunotherapy, such as miRNA, which could affect PD-L1 expression and their function in a strictly defined manner, and possibly simultaneously provide supportive treatment, affecting them in a strictly defined manner. Potential molecules that we initially tested under this account were the miR-200, miR-1184, and miR-1255a families. We have proved that carlina oxide modulates the expression of all miRNAs with a simultaneous change in PD-L1 mRNA level. In subsequent studies, an experiment on the synergistic effect of carlina oxide at the appropriate dose (most likely 50 µg/mL) and anti-PD-L1 antibodies on apoptosis of cancer cells should be carried out.

Carlina oxide exhibited high in vivo toxicity with LC_50_ = 10.13 µg/mL within the 96 h of exposure in the ZFET test. Motherwort (*Leonurus japonicus*) essential oil displayed LC_50_ of ~10 µg/mL and was demonstrated to be teratogenic to zebrafish embryos [17]. These LC_50_ values are much lower than those observed for morphine (LC_50_ = 9915.1 µg/mL) or atropine (607.8 µg/mL), but comparable to LC_50_ values of well-established toxins like nicotine (35.1 µg/mL) and strychnine (20.8 µg/mL) [18]. Coagulation of embryos was the major lethal effect of carlina oxide, indicating that this drug most prominently affects early stages of organism development. Moreover, malformations and bradycardia observed at the 96 h of the test in larval zebrafish subjected to carlina oxide show that this compound exhibits teratogenic and cardiodepressive potential in living systems.

## 4. Conclusions

Here, we demonstrated that carlina oxide displays toxic effects to cells in culture and to living organisms. Collected data support the hypothesis that carlina oxide is responsible for toxic effects of *C. acaulis* root extracts. The data indicate that *C. acaulis*-based extracts considered for therapeutic use should be deprived of carlina oxide in total.

## 5. Materials and Methods

### 5.1. Plant Material

The *C. acaulis* L. plants were obtained from the Botanical Garden of Maria Curie-Skłodowska University (UMCS) in Lublin, Poland, identified by Mykhaylo Chernetskyy, and deposited in the Botanical Garden of UMCS (voucher specimen no. 2005A). They were grown in an open field. The plants were collected in the second half of July 2018 in the vegetative stage. Once harvested, whole roots were thoroughly washed with tap and distilled water, and then they were dried at room temperature.

### 5.2. Isolation, Investigation of the Identity and Purity of Carlina Oxide

Carlina oxide was obtained by distillation of *C. acaulis* roots in the Deryng apparatus. The purity of the standard was evaluated using GC-MS, and the identity was confirmed by IR, Raman, and NMR spectroscopy according to the methodology published previously [19]. GC-MS analysis was performed using the Agilent GC-MSD system with a DB-5MS capillary column (30 m × 0.25 mm, 0.25 µm film thickness (Agilent Technologies, Santa Clara, CA, USA), and a split-splitless injector. The temperature was programmed from 70 °C to 290 °C at a rate of 5 °C/minute, and then held for 10 min at 290 °C. The temperature of the injector and interface was 250 °C and 300 °C, respectively. The injection volume was 1 µL of 1% hexane solution (split ratio 1:50). Helium was used as a carrier gas at a flow rate of 1 mL/min. The analysis was conducted with the use of a quadrupole mass spectrometer Agilent 5975 (Agilent Technologies, Santa Clara, CA, USA) and performed in the electron ionization mode at 70 eV, scan time 1s, where the acquisition mode was a full scan (40–550 m/z). Identification of the analytes was based on a comparison of their retention time with their linear indices relative to a series of n-alkanes (C8-C20). Mass spectra were interpreted based on a previous publication [19] and mass spectra library of National Institute of Standards and Technology (NIST) resources. The ^1^H NMR and ^13^C NMR spectra were recorded on a Bruker Avance 600 spectrometer (Bruker BioSpin GmbH, Rheinstetten, Germany) in DMSO-*d*_6_. The Raman spectra were recorded using a Thermo Scientific DXR confocal Raman Microscope equipped with Omnic 8 software (Thermo Fisher Scientific, Madison, WI, USA). ATR-IR spectra were recorded using a Thermo Scientific Nicolet 6700 FTIR spectrophotometer equipped with an ATR attachment with diamond crystal and Omnic software version 8.2. All spectroscopic measurements were made according to methodologies published previously [19].

### 5.3. Cell Culture

Human melanoma cell lines UACC-647 (RRID:CVCL_4049), and UACC-903 (RRID:CVCL_4052) were a generous gift from Michel Bernier (National Institute on Aging, National Institutes of Health, Baltimore, MD, United States). The C32 (RRID:CVCL_1097) melanoma cell line was obtained from the American Type Culture Collection (ATCC, Manassas, VA, United States; CRL-1585). Human BJ fibroblasts (RRID:CVCL_3653) were also obtained from ATCC (CRL-2522). Cell culture was conducted according to methodology published previously [10].

### 5.4. Cell Apoptosis

The cells were seeded in complete medium (10% FBS) in a 6-well plate at the density of 1.25 × 10^5^ (for C32 line), 2 × 10^5^ (for UACC-647 line), 3 × 10^5^ (UACC-903 line), and 2 × 10^5^ cells per well (for the BJ fibroblasts line). After 24 h, the supernatant was aspirated and the carlina oxide solution were added at the following concentrations: 50, 25, 12.5, 6.25, and 3.125 μg/mL. Vehicle (EtOH) was used as a control. The cultures were incubated for another 24 h, and then the cells were stained with Intracellular Caspase Detection ApoStat (R&D System) directly during the last 30 min of culture. Ten microliters of ApoStat per 1 mL of the culture volume was added and incubated at 37 °C. Unstained cells served as a negative control. After the staining period, the cells were harvested, centrifuged at 500× *g* for 5 min, and washed once with 4 mL of phosphate-buffered saline (PBS) to remove unbound reagent. The cells were resuspended in 500 µL of PBS and immediately analyzed by flow cytometry.

### 5.5. Western Blotting

The UACC-647 cells were seeded out onto 6-well plates and cultured overnight to achieve confluency of ca. 80%. On the next day, the cells were subjected to either vehicle (EtOH, 0.1%) or increasing concentrations of carlina oxide (3.125, 25, 50 µg/mL) for 24 h. Then, the cells were washed in ice-cold DPBS and lysed for 10 min in 200 µL of lysis buffer from Cell Signaling Technology. The lysates were centrifugated for 10 min at 18,100× g at 4 °C. Protein concertation was assessed in obtained supernatants using a bicinchoninic acid assay. Then, equal amounts of protein were loaded onto the polyacrylamide gels and subjected to electrophoresis. Separated proteins were transferred onto a PVDF membrane using an iBlot2 device. The membrane was blocked with 3% milk in TBST for 30 min and subsequently subjected to primary antibodies (16 h of incubation with constant agitation at 4 °C). Primary antibodies from Cell Signaling Technology raised against the following targets were used: AKT (#9272), ERK1/2, (#4695), eEF2 (#2332), and PCNA (#13110). β-Actin served as loading control (abcam, #ab6276). Unbound antibodies were washed two times for 4 min and two times for 2 min in TBST. Then, the membranes were exposed to either HRP-linked or Alexa Fluor 555-conjugated secondary antibodies for 30 min. Unbound secondary antibodies were washed using the same washing protocol as before. When necessary, the membranes were subjected to Westar Supernova ECL reagent (Cyanagen) for 2 min. The immunoreactive bands were detected using an Azure 400 Western blot imaging system (Azure Biosystems) and quantified densitometrically using ImageJ 1.52p.

### 5.6. Carlina Oxide and MicroRna Expression in Cell Lines

#### 5.6.1. RNA Isolation

Isolation of total RNA with the microRNA fraction was performed using the miRNeasy Mini Kit (Qiagen, Germany) according to the manufacturer’s instruction. Isolation was carried out from cells of an amount of 1 × 10^4^ scratched from a culture plate. RNA was stored in −80 °C until cDNA synthesis was performed.

#### 5.6.2. Assessment of PD-L1 (mRNA) Expression

First, we performed reverse transcription-polymerase chain reaction (RT-PCR) to obtain cDNA. RT-PCR was conducted using a high-capacity RNA-to-cDNA kit (Life Technologies, Carlsbad, CA, USA) according to the manufacturer’s instructions. Next, we measured PD-L1 relative expression in reference to GAPDH and β-actin as internal controls. Quantitative PCR (qPCR) was performed using TaqMan Fast Advanced Master Mix and TaqMan Gene Expression Assay (Life Technologies, Carlsbad, CA, USA) on an Illumina Eco Real-Time PCR System (Illumina, San Diego, CA, USA). Twenty microliters of PCR reaction contained 10 µL of TaqMan Fast Advanced Master Mix, 1 µL of TaqMan Gene Expression Assay (PD-L1 or reference assay), 5 µL of RNase-free water, and 4 µL of cDNA. In a negative control reaction, nuclease-free water was used instead of cDNA. Reaction was carried out in the following conditions: 95 °C for 20 s, 40 cycles for 95 °C for 3 s, and 62 °C for 30 s. Analysis was performed using the 2^−ΔCt^ method.

#### 5.6.3. Assessment of miRNA Witch Ability to Regulate Pd-l1 Expression

We analyzed the expression of six microRNAs (selected by TargetScan 7.0 tool) complementary to 3′UTR (untranslated region) of PD-L1 mRNA: miR-141-3p, miR-200a-3p, miR-200b-3p, miR-200c-3p, miR-1184, and miR-1255a. First, RT-PCR was conducted for miRNA using a TaqMan Advanced miRNA cDNA Synthesis Kit (Life Technologies) according to the manufacturer’s instructions. Real-time PCR was performed on an Illumina Eco Real-Time PCR System (Illumina Inc). Twenty microliters 20 µL of PCR mixture contained: 10 µL of TaqMan Fast Advanced Master Mix, 1 µL of TaqMan Fast Advanced miRNA Assay, 4 µL of RNase-free water, and 5 µL of cDNA. As an internal control, miRNA-191 was used. In a negative control reaction, nuclease-free water was used instead of cDNA. The reaction was conducted in the following conditions: 95 °C for 20 s, 40 cycles at 95 °C for 5 s, and 60 °C for 30 s. Analysis was performed using 2^−ΔCt^.

### 5.7. Zebrafish Embryo Acute Toxicity (ZFET) Test

The ZFET test was used to determine in vivo acute toxicity of carline oxide. AB line zebrafish eggs were purchased from the Centre of Experimental Medicine, Medical University of Lublin, Poland. The test procedure was established on the basis of OECD Guidelines for the Testing of Chemicals, Test No. 236 [20] with further modifications in accordance with Nishimura et al. [21]. Briefly, 24 embryos per concentration were exposed to carlina oxide (3.125, 4.688, 6.25, 9.375, 12.5, 18.75, and 25 μg/mL), E3 medium (negative control), 0.5% DMSO (solvent control), or 20 µg/mL acetone (positive control). Exposure started at the 16-cell stage and was continued for a period of 96 h. Embryos (one embryo per well) were allocated across 96-well plates and maintained in the incubator with a controlled temperature (28.5 ± 0.5 °C) and day–night cycle (14 h light/10 h dark). The solutions were renewed every 24 h at a volume of 150 μL per well. Apical observations of acute toxicity were made every 24 h. The presence of the following toxic effects was verified: (1) coagulation of the embryo, (2) lack of somite formation, (3) non-detachment of the tail, and (4) a lack of heartbeat. At the end of the test, acute toxicity was determined based on a positive outcome in any of the four apical observations recorded. In order to ensure equal exposure to the tested drug, all larvae were dechorionated at the first observation time point, at which larvae subjected to carlina oxide started to hatch, that is, after 72 h of the test. Additionally, the heart rates in 96 hpf larval zebrafish (10 larvae per concentration, randomly chosen) was measured by counting the beats in 15 s and extrapolating it to beats per minute.

### 5.8. Statistics

The median lethal dose (LC_50_) was determined by nonlinear, four parameters regression analysis. The D’Agostino and Pearson normality test was used to verify the normality of the data. Bartlett’s test was employed to assess whether standard deviations were statistically different among samples. The results were processed either by the one-way analysis of variance (one-way ANOVA) or by two-way ANOVA. This was followed by Dunnett’s post hoc test. The confidence limit of *p* < 0.05 was considered statistically significant. All statistical analyses were performed using GraphPad Prism 7.

## Figures and Tables

**Figure 1 toxins-12-00239-f001:**
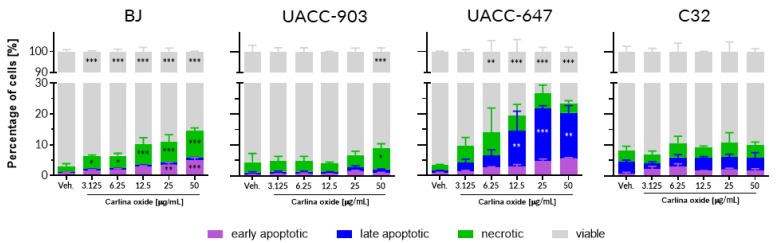
Carlina oxide elicits cell-line-specific toxicity. Percentages of apoptotic, late apoptotic, and necrotic cells upon treatment with increasing concentrations of carlina oxide. Statistical analysis: two-way ANOVA with Dunnett’s post-hoc test; *, *p* < 0.05; **, *p* < 0.01; ***, *p* < 0.001. Data originates from three independent experiments.

**Figure 2 toxins-12-00239-f002:**
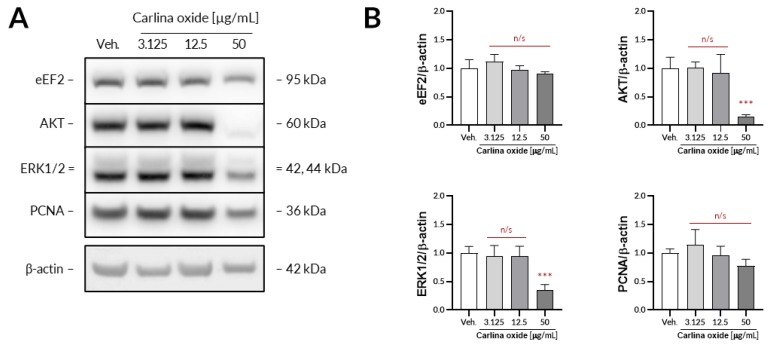
Carlina oxide affects the expression of key signaling nodes in UACC-647 cells. (**A**) Representative Western blots obtained based on UACC-647 cells subjected to increasing concentrations of carlina oxide (3.125, 12.5, and 50 µg/mL) for 24 h. (**B**) Densitometric analysis of the expression of eukaryotic elongation factor 2 (eEF2), AKT, extracellular signal-regulated kinase 1/2 (ERK1/2), and proliferating cell nuclear antigen (PCNA). Statistical analysis: one-way ANOVA with Dunnett’s post hoc test; ***, *p* < 0.001; n/s, not significant. Data originated from four independent experiments.

**Figure 3 toxins-12-00239-f003:**
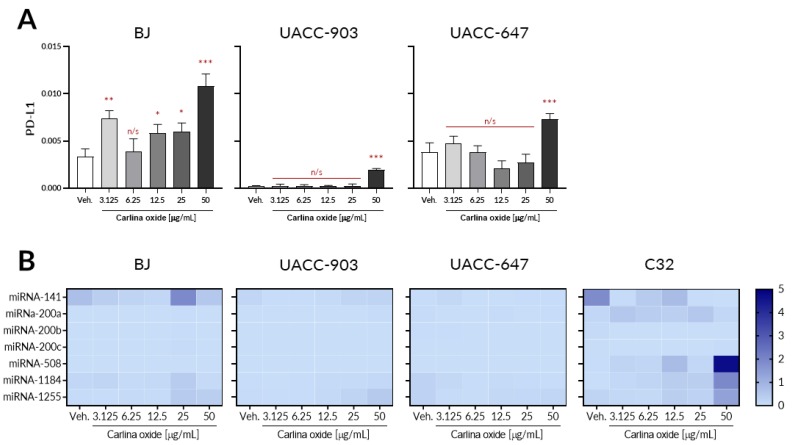
Carlina oxide affects PD-L1 expression. (**A**) Expression of PD-L1 was studied in BJ fibroblasts and UACC-903, UACC-647, and C32 melanoma cells. The cells were exposed to a range of carlina oxide concentrations (3.125–50 µg/mL) and PD-L1 expression was assessed by qPCR. There was no detectable expression of PD-L1 in C32 cells. One-way ANOVA was conducted with Dunnett’s post hoc test; ***, *p* < 0.001; n/s, not significant. (**B**) Expression of miRNA previously linked with PD-L1 expression was studied in the same set of cell lines by qPCR. Obtained expression values (fold-change) are depicted as heatmap. Data originate from three independent experiments.

**Figure 4 toxins-12-00239-f004:**
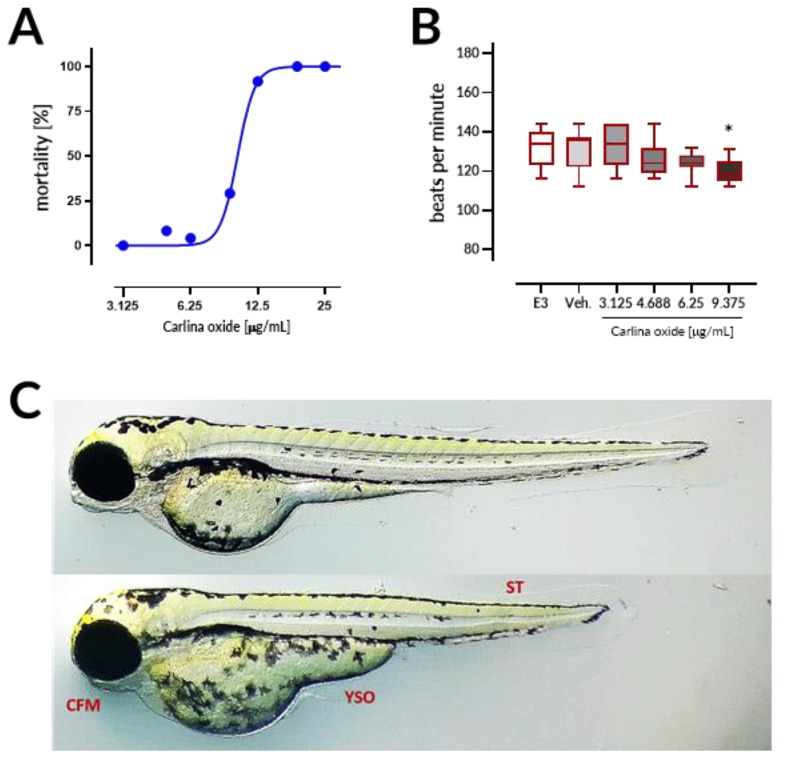
Carlina oxide is toxic to zebrafish embryos. (**A**) Mortality of zebrafish treated with carlina oxide (3.125–25 μg/mL) at the 96 h of the ZFET test. (**B**) Effects of carlina oxide on the heart rate of larval zebrafish at the 96 hpf of the ZFET test. Data represent mean ± SD, n = 10, *, *p* < 0.05 vs. negative control (E3), Dunnett’s test. (**C**) Teratogenic effects of carlina oxide on zebrafish larvae at the 96 h of the ZFET test (top panel: negative control – E3 medium; bottom panel: carlina oxide, 25 μg/mL). CFM, craniofacial malformation; YSO, yolk sac oedema; ST: shortened tail.

**Table 1 toxins-12-00239-t001:** The number of dead embryos/larvae exposed to carlina oxide in the zebrafish acute toxicity test ZFET test.

Time Point (Hours)	Carlina Oxide (μg/mL)						
	3.125	4.688	6.25	9.375	12.5	18.75	25
24/48/72	0	2	1	5	22	24	24
96	0	2	1	7	22	24	24

The number of dead embryos/larvae was constant at the 24, 48, and 72 h of apical observations in the ZFET test. All dead embryos were recorded due to coagulation. Two more larvae were found dead in the 96 h time point at a concentration of 9.375 µg/mL due to the lack of heartbeat.

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
