# Peer review of "Toxicity of Carlina Oxide—A Natural Polyacetylene from the *Carlina acaulis* Roots—In Vitro and in Vivo Study"

_toxins, 2020, doi:10.3390/toxins12040239_

Round 1

Reviewer 1 Report

The authors report on the investigation of an important chemical in root extracts of Carlina acaulis. The authors overplay the lack of study of these materials a bit. This is heavily reflected in the discussion section. It seems to be by journal guidelines to order the sections as presented in the manuscript. However, a discussion without any relation to other studies is insufficient. At the current stage, this will lead to my recommendation of major rewrite. A simple internet search for the compound reveals quite a number of studies in the arena of these materials. The current data do need to be related to those reports – what is novel, what is different, where does that lead?

Stay in past tense in your descriptions. Include critical information on the replication numbers and reproducibility of results.

More specific comments:

Ln 6-7: treatment measures for what? This is far to vague

Ln 13: add: “.. in vivo in a zebra fish accute toxicity test (ZFET)..”

Ln 14: “..tested lines. Specific to cell lines, apoptosis and necrosis were measured.” Omit the “However..” sentence.

Ln 15: what is AKT?

Ln 18: this is not particularly low. The authors don’t use the chance in their discussion to discuss this value. If they could convincingly illustrate that this is low compared to other materials….

Ln 21: no data on essential oils in general – no justification for this comment.

Ln 30: In proper writing, “however” is only used for stark and unexpected contrasts. That is not the case here how about “Perhaps ts importance was lost because…”

Ln 41 “..we hypothesized..”

Ln 48: “.. assessment of carlina oxide…”

Ln 50: what is the origin? How are those cell lines hypothetically different?

Ln 52: “.. decreased at a dose of..”

Ln126 “..contains..”

Ln190: really 500 ml?

Author Response

Dear Editors and Reviewers,

Thank you for your letter and for the Reviewers’ comments concerning our manuscript entitled „Toxicity of carlina oxide – a natural polyacetylene from the Carlina acaulis roots. In vitro and in vivo study”. Those comments were valuable and helped us to revise and improve our paper. We have studied comments carefully and have made correction which we hope meet with approval. Revised portions are marked in color in the paper. Once again, thank you very much for your comments and suggestions.

Reviewer 1

Q1.1: The authors report on the investigation of an important chemical in root extracts of Carlina acaulis. The authors overplay the lack of study of these materials a bit. This is heavily reflected in the discussion section. It seems to be by journal guidelines to order the sections as presented in the manuscript. However, a discussion without any relation to other studies is insufficient. At the current stage, this will lead to my recommendation of major rewrite. A simple internet search for the compound reveals quite a number of studies in the arena of these materials. The current data do need to be related to those reports – what is novel, what is different, where does that lead?

Authors response: We are very grateful to the reviewer for the comment regarding the discussion of our finding in relation to other investigations. It has prompted us to improve the discussion section of the manuscript. We compared our results generated using BJ fibroblasts with the data obtained by Pavela and colleagues in NHF-A12 fibroblasts. Cytotoxicity generated by carlina oxide in both cell lines was comparable. Please, see the Discussion section for more details.

Q1.2: Stay in past tense in your descriptions. Include critical information on the replication numbers and reproducibility of results.

Authors response: Number of replicates was included in figures’ descriptions, as advised by the reviewer. Some sentences were converted to past tense, for the sake of clarity and consistency.

Q1.3: Ln 6-7: treatment measures for what? This is far to vague

Authors response: Thank you for Your comment. We have modified the sentence: There are several reports indicating that the roots of the Carlina acaulis L. used to be commonly applied as a treatment measure in skin diseases and as an antiparasitic agent from antiquity to XIX century; however, nowadays it has lost its importance.

Q1.4: Ln 13: add: “.. in vivo in a zebra fish acute toxicity test (ZFET)..”

Authors response: The abstract was modified according to the reviewer’s suggestion.

Q1.5: Ln 14: “..tested lines. Specific to cell lines, apoptosis and necrosis were measured.” Omit the “However..” sentence.

Authors response: The abstract was modified according to the reviewer’s suggestion.

Q1.6: Ln 15: what is AKT?

Authors response: AKT is a cellular kinase that constitutes one of the key signaling nodes involved in the control of cell proliferation and metabolism. It is also referred to as PKB (protein kinase B). The abstract was modified to include the function of AKT protein.

Q1.7: Ln 18: this is not particularly low. The authors don’t use the chance in their discussion to discuss this value. If they could convincingly illustrate that this is low compared to other materials….

Authors response: Encouraged by the Reviewer, we expanded the discussion regarding the LD50 of carlina oxide in zebrafish embryos. We compared it LD50 of essential oil from Leonurus japonicus and of some well-established toxins as nicotine or strychnine. Please, see the revised Discussion section for more details.

Q1.8: Ln 21: no data on essential oils in general – no justification for this comment.

Authors response: Carlina oxide is the main compound in C. acaulis root oil, but the phrase we use can be misleading. Thank you for your right consideration. We've modified this sentence. This remark was also taken into account in the conclusions section.

Q1.9:  Ln 30: In proper writing, “however” is only used for stark and unexpected contrasts. That is not the case here how about “Perhaps its importance was lost because…”

Authors response: The sentence was modified according to the reviewer’s suggestion.

Q1.10: Ln 41 “..we hypothesized..”

Authors response: The error was corrected.

Q1.11: Ln 48: “.. assessment of carlina oxide…”

Authors response: The error was corrected.

Q1.12: Ln 50: what is the origin? How are those cell lines hypothetically different?

Authors response: The issue of differences between the melanoma cell lines employed in our study was dressed in the Discussion section. Bittner and colleagues characterized a large panel of melanoma cell lines and discovered that they display significant heterogenicity in terms of gene expression, mutations, motility, invasiveness, and vasculogenic mimicry.

Q1.13: Ln 52: “.. decreased at a dose of..”

Authors response: The error was corrected.

Q1.14: Ln126 “..contains..”

Authors response: The error was corrected.

Q1.15: Ln190: really 500 ml?

Authors response: The error was corrected.

Reviewer 2 Report

The study entitled” Toxicity of Carlina Oxide – a Natural Polyacetylene from the Carlina acaulis Roots. In vitro and in vivo Study assessed the toxicity of C. acaulis-derived carlina oxide in vitro and in vivo models to address the safety of C. acaulis root preparations used in folk medicine and evaluate the possibility of reintroducing this plant into phytotherapy. I think that this manuscript very important in medicines and toxicology to approve or disapprove the toxicity of the plant or it constituents.

My suggestions are

  1. Abstract: There are several reports indicating that the roots of the Carlina acaulis used to be commonly applied as a treatment measure from ancient to XIX century; however, nowadays it has lost its importance. Rephrase kindly to be clear for readers.
  2. In the method section of the abstract part Add kindly how the name of isolation and identification methods of carlina oxide in your experiment.

3. In the methodology section kindly indicate in your manuscript the positive and negative controls that you used.

4.Add subsection (plant material) explaining from where and when you collected Carlina acaulis In addition, who identified the plant and its voucher specimen code. Moreover, how you dried the roots and the drying conditions.

5. Add subsection (GC-MS) and IR, Raman, and NMR spectroscopy used methods and under which conditions.

6. In the results section add the chemical part of your work (GC-MS), IR and NMR spectral charts (In the supplementary file and their characterizations in the results under the section spectral analysis.

7. Major grammatical, typos and editing corrections required to improve your manuscript to be clear for readers.

8. Minor required corrections included in the attached file

Author Response

Dear Editors and Reviewers,

Thank you for your letter and for the Reviewers’ comments concerning our manuscript entitled „Toxicity of carlina oxide – a natural polyacetylene from the Carlina acaulis roots. In vitro and in vivo study”. Those comments were valuable and helped us to revise and improve our paper. We have studied comments carefully and have made correction which we hope meet with approval. Revised portions are marked in color in the paper. Once again, thank you very much for your comments and suggestions.

Reviewer 2

Q2.1: The study entitled “Toxicity of Carlina Oxide – a Natural Polyacetylene from the Carlina acaulis Roots. In vitro and in vivo Study” assessed the toxicity of C. acaulis-derived carlina oxide in vitro and in vivo models to address the safety of C. acaulis root preparations used in folk medicine and evaluate the possibility of reintroducing this plant into phytotherapy. I think that this manuscript very important in medicines and toxicology to approve or disapprove the toxicity of the plant or its constituents.

Authors response: Dear Reviewer, thank you very much for acknowledging our efforts. We hope that our corrections will be satisfactory.

Q2.2: Abstract: There are several reports indicating that the roots of the Carlina acaulis used to be commonly applied as a treatment measure from ancient to XIX century; however, nowadays it has lost its importance. Rephrase kindly to be clear for readers.

Authors response: Thank you for Your comment. We have modified the sentence: There are several reports indicating that the roots of the Carlina acaulis L. used to be commonly applied as a treatment measure in skin diseases and as an antiparasitic agent starting from antiquity to XIX century; however, nowadays it has lost its importance.

Q2.3: In the method section of the abstract part add kindly how the name of isolation and identification methods of carlina oxide in your experiment.

T Authors response: thank you very much for your valuable suggestion. Information has been added. Please, see new abstract and new section 5.2 for updated isolation and identification methodology.

Q2.4: In the methodology section kindly indicate in your manuscript the positive and negative controls that you used.

Authors response: Requested pieces of information regarding the applied experimental controls were added throughout the Materials and methods section. We explained that “unstained cells served as a negative control” in cell apoptosis assessment (see new section 5.4 for details). We also clarified that “nuclease free water was used instead of cDNA” in negative controls when PD-L1 and miRNA expression was studied (see new section 5.6.2 and 5.6.3 for details).

Q2.5: Add subsection (plant material) explaining from where and when you collected Carlina acaulis. In addition, who identified the plant and its voucher specimen code. Moreover, how you dried the roots and the drying conditions.

Authors response: This is an extremely vital point. We have added a subsection aimed to specifically address these recommendations. Please see new section 5.1 for detailed information on collection, identification and drying of the plant material.

Q2.6: Add subsection (GC-MS) and IR, Raman, and NMR spectroscopy used methods and under which conditions.

Authors response: Dear reviewer, we have added basic data on chemical analysis. Detailed data on the identification and determination of carlin oxide were presented in our previous publications. However, we agree that for full transparency of the manuscript and scientific solidity they should be included in the text.

Q2.7: In the results section add the chemical part of your work (GC-MS), IR and NMR spectral charts (In the supplementary file and their characterizations in the results under the section spectral analysis.

Authors response: Dear Reviewer, we have added a short subsection in the results section and chromatogram and spectra in the supplementary material. Detailed spectrum analysis is the subject of our previous publication. In addition, the composition of the C. acaulis root oil is well described by other authors.

Q2.8: Major grammatical, typos and editing corrections required to improve your manuscript to be clear for readers.

Authors response: English throughout the manuscript has been corrected.

Q2.9: Minor required corrections included in the attached file                         

Authors response: Dear Reviewer, we are very grateful for such a detailed review. We have made every effort to make appropriate corrections.

Round 2

Reviewer 1 Report

The authors have addressed the concerns to my satisfaction. Thank you for doing this. I believe the manuscript is much improved because of those additions. From my perspective, it can be accepted as it is now. Thank you.

Reviewer 2 Report

The authors improved the quality of their manuscript and established all the required corrections but still needs moderate grammatical, typos and editing corrections